# The Identification of Subphenotypes and Associations with Health Outcomes in Patients with Opioid-Related Emergency Department Encounters Using Latent Class Analysis

**DOI:** 10.3390/ijerph19148882

**Published:** 2022-07-21

**Authors:** Neeraj Chhabra, Dale L. Smith, Caitlin M. Maloney, Joseph Archer, Brihat Sharma, Hale M. Thompson, Majid Afshar, Niranjan S. Karnik

**Affiliations:** 1Division of Medical Toxicology, Department of Emergency Medicine, Cook County Health, Chicago, IL 60612, USA; 2Department of Emergency Medicine, Rush Medical College, Rush University, Chicago, IL 60612, USA; 3Addiction Data Science Laboratory, Department of Psychiatry & Behavioral Science, Rush University Medical Center, Chicago, IL 60612, USA; dale_smith@rush.edu (D.L.S.); brihat_sharma@rush.edu (B.S.); hale_thompson@rush.edu (H.M.T.); nkarnik@uic.edu (N.S.K.); 4Department of Psychology, Olivet Nazarene University, Bourbonnais, IL 60914, USA; 5Doctor of Medicine Program, Rush Medical College, Rush University, Chicago, IL 60612, USA; caitlin_m_maloney@rush.edu; 6School of Medicine and Public Health, University of Wisconsin, Madison, WI 53715, USA; joseph_archer@rush.edu; 7Department of Medicine, University of Wisconsin-Madison, Madison, WI 53715, USA; mafshar@medicine.wisc.edu; 8Institute for Juvenile Research, Department of Psychiatry, University of Illinois Chicago, Chicago, IL 60612, USA

**Keywords:** opioid misuse, emergency department, latent class analysis, opioid epidemic

## Abstract

The emergency department (ED) is a critical setting for the treatment of patients with opioid misuse. Detecting relevant clinical profiles allows for tailored treatment approaches. We sought to identify and characterize subphenotypes of ED patients with opioid-related encounters. A latent class analysis was conducted using 14,057,302 opioid-related encounters from 2016 through 2017 using the National Emergency Department Sample (NEDS), the largest all-payer ED database in the United States. The optimal model was determined by face validity and information criteria-based metrics. A three-step approach assessed class structure, assigned individuals to classes, and examined characteristics between classes. Class associations were determined for hospitalization, in-hospital death, and ED charges. The final five-class model consisted of the following subphenotypes: Chronic pain (class 1); Alcohol use (class 2); Depression and pain (class 3); Psychosis, liver disease, and polysubstance use (class 4); and Pregnancy (class 5). Using class 1 as the reference, the greatest odds for hospitalization occurred in classes 3 and 4 (Ors 5.24 and 5.33, *p* < 0.001) and for in-hospital death in class 4 (OR 3.44, *p* < 0.001). Median ED charges ranged from USD 2177 (class 1) to USD 2881 (class 4). These subphenotypes provide a basis for examining patient-tailored approaches for this patient population.

## 1. Introduction

Drug overdose is a leading cause of death in the United States, with the majority involving an opioid [1]. There were approximately 50,000 opioid overdose deaths reported in 2019, and the recent COVID-19 pandemic has seen increases in opioid-related deaths [2,3]. Patients with opioid misuse disproportionately utilize emergency medical services, with almost 1.5 million opioid-related emergency department (ED) encounters annually [4,5,6]. ED patients have been found to have more severe substance use patterns compared with patients seen in the primary care setting [7]. The one-year mortality rate for patients following an opioid-related ED visit may be greater than 5% [8,9]. Therefore, the ED represents a critical point of contact between patients with opioid misuse and the health care system as ED encounters are opportunities for interventions to prevent future morbidity and mortality [10]. Several interventions initiated in the ED setting have been effective in improving outpatient treatment engagement and preventing overdose, including buprenorphine initiation, take-home naloxone, team-based care coordination, and peer recovery support services [11,12,13,14,15,16]. While the utilization and expansion of these services have improved the care of ED patients with opioid misuse, none have been shown to be uniformly effective.

Opioid misuse is characterized by a heterogeneous pattern influenced by several factors including type of opioid used, severity of illness, exposure route, comorbid conditions, and social determinants of health [17]. In short, opioid misuse is not a one-size-fits-all condition. It is therefore unsurprising that ED-based interventions have not been uniformly effective for all patients. Current evidence suggests that treatment responses and clinical outcomes differ between specific subphenotypes of patients with opioid misuse [18,19]. Such subphenotypes are defined by varying risks for poor outcome or shared underlying biologic factors. Ideally, management approaches for patients with opioid-related encounters would be tailored to the needs of individual patients. To personalize treatment approaches within the ED setting, clinically relevant subphenotypes of patients with opioid misuse must first be discovered and described. Distinct clusters of patients with opioid misuse, defined by shared characteristics, have been identified in the general population and clinical settings outside of the ED [20,21,22]. There is a need to determine if such subphenotypes exist within the ED setting, and if these subphenotypes differ in patient characteristics and outcomes. These subphenotypes would inform treatment considerations, resource allocation, public policy, and future research.

To identify and describe subphenotypes of patients with opioid-related ED encounters, we applied latent class analysis (LCA) to a nationally representative sample of ED encounters. LCA utilizes multivariate categorical data to surface informative patterns in a dataset to cluster subsets of patient encounters based on shared characteristics [23,24]. LCA has been used to detect clinically informative subphenotypes of patients for health conditions including alcohol misuse, e-cigarette use, and respiratory failure, among others [25,26,27]. We hypothesize that by using LCA we will detect distinct and clinically relevant subphenotypes of ED patients with opioid-related encounters; these subphenotypes signal key differences in patient characteristics with implications for patient outcomes as well as potential treatment and intervention pathways.

## 2. Materials and Methods

### 2.1. Data Source

Clinical data were obtained from the National Emergency Department Sample (NEDS), the largest all-payer emergency department database in the United States. NEDS is a nationally representative sample developed and maintained by the Healthcare Cost and Utilization Project (HCUP) [28]. NEDS contains systematically stratified discharge data from approximately 1000 contributing hospitals from all regions of the United States, using 20% sampling across strata (geographic region, urban or rural location, teaching status, ownership, and trauma-level designation) from all emergency department visits nationwide. The database contains a combination of structured billing and demographic variables.

### 2.2. Study Population

The sample population included patients over 12 years of age with an opioid-related International Classification of Diseases, Tenth Revision (ICD-10) diagnosis code (Appendix A) as a primary or secondary diagnosis excluding codes negating opioid use, indicating remote use, or indicating disease in remission. Opioid-related ICD-10 codes were adopted from the Healthcare Cost and Utilization Project and represent a commonly used definition for health services research [29]. ICD-10 codes reflect final billing diagnosis codes used for claims by payers. For the purposes of the study, the sample was not stratified by encounter reason, such as opioid overdose or withdrawal, as the goal of the investigation was to identify subphenotypes by patient characteristics rather than encounter-specific characteristics, which are more likely to change over time. Adolescent encounters were included as substance misuse because such use often begins in the adolescent period, and many characteristics of substance use do not greatly differ between adolescents and adults [30]. NEDS does not utilize patient identifiers, so multiple encounters by the same patient were included as independent observations. We merged NEDS files from 2016 and 2017 for analysis. This time frame was chosen due to the increasing contribution of synthetic opioids to the opioid epidemic in recent years and for data consistency as 2016 represents the first entire year that included ICD-10 diagnosis codes, covariates, and sample weights. The year 2017 is the most recent year with available data at the time of analysis.

### 2.3. Variables and Covariates

Candidate variables within NEDS included those with previously established associations with opioid misuse, sufficient interpretability for providers, and conceptual association with overdose risk [20,21]. These class-defining variables for the LCA models consisted of groups of diagnosis codes for the following disease processes: (1) chronic pain; (2) alcohol use disorders; (3) psychoses; (4) depression; (5) liver disease; (6) pregnancy; (7) cocaine use; and (8) amphetamine use (Appendix A). These variables were chosen a priori based on their previous use in effectively clustering patients with substance misuse using LCA in settings outside the ED [20,21,31]. Demographic variables utilized as covariates included age, sex, payer, hospital urban–rural designation, and median household income for the patient’s ZIP code (stratified by quartile); these variables were selected because of their known associations with substance use [32,33]. The six-category urban–rural classification scheme was developed by the National Center for Health Statistics to differentiate between central and fringe metropolitan areas with smaller areas subdivided by population. Race, ethnicity, and level of education are not variables collected by NEDS and were therefore not utilized as covariates.

### 2.4. Statistical Analyses

A series of nested LCA models with K + 1 classes were iteratively fit up to a 10-class model utilizing the dichotomous variables derived from ICD-10 codes and weights accounting for the complex sampling design of NEDS. Models were fit by maximum likelihood estimation using the expectation–maximization algorithm. The optimal number of latent classes was determined using clinical interpretability in addition to the exploration of information criteria-based metrics and model entropy. The goal was to identify the minimum number of classes that best described and fit the data. Among models with similar performance, the most parsimonious model was chosen. The metrics for model selection included the adjusted Bayesian information criterion (aBIC), Akaike information criterion (AIC), and chi-squared goodness of fit. Initial LCA was explored on all patients across three years (2015–2017), although because covariate information and sample weights were not available for 2015, weighted analyses were conducted only with the 2016 and 2017 data after verifying the robustness of class structure within this reduced sample.

Class sizes were determined by class membership using the highest posterior probability for each patient encounter to identify class sizes that were either a too-large or too-small proportion of the population to be clinically informative. Previous simulations have shown that classes accounting for less than 5% of the total cohort are less likely to be informative [34]. Face validity of the subphenotypes was determined by clinical interpretability on manual review by researchers with expertise in emergency care, substance use research, and/or LCA methodology (NC, MA, CM, and NK).

Covariates were explored using the three-step approach due to recent concerns regarding utilizing covariates in the initial exploration of class structure increasing the potential for bias [35,36]. The three-step approach initially assesses class structure independent of covariates, assigns individuals to classes based on posterior probability of class membership, and then explores the covariates’ relationships with class membership. We also explored the outcomes of inpatient hospitalization, in-hospital death, and ED charges (USD) across the latent classes. To identify differences in demographics and outcomes between subphenotypes, descriptive statistics and odds ratios were calculated for the demographic variables and the patient outcomes. Odds ratios were determined by multinomial logistic regression predicting class membership by covariate while correcting for classification error. They represent raw associations between covariate and class and were not adjusted for other covariates used in the latent class analysis. Due to the large sample size, traditional significance tests and *p*-values were uninformative (all *p*-values < 0.001) and standard errors were generally < 0.001. Emphasis was thus placed on the descriptive statistics and odds ratios for class membership or class makeup based on covariates. For interpretability, the outcome of ED charges was determined as median raw charges for each class, and for between-class comparisons, ED charges were transformed to a natural logarithmic scale to account for the strong positive skew and non-normal distribution. Analyses were performed using Stata Statistical Software: Release 17 (College Station, TX, USA: StataCorp LLC. StataCorp. 2019) and Mplus (Los Angeles, CA: Muthén & Muthén). Missingness occurred mostly in the ED charge field, a limitation noted by HCUP for the 2016 dataset, and was handled by listwise deletion [28]. The protocol was reviewed and deemed exempt by the institutional review board of Rush University Medical Center. This study conforms, where applicable, to the Strengthening the Reporting of Observational Studies in Epidemiology (STROBE) statement guidelines (Appendix A) [37].

## 3. Results

### 3.1. Model Selection

A total of 14,057,302 ED encounters met the inclusion criteria for analysis. The information criteria and smallest class sizes from LCA models with classes from 1 to 10 are shown in Table 1. All models with six or more classes contained multiple classes representing less than 5% of the study population. Information criteria generally indicated preference for larger numbers of classes (Appendix A); however, the five-class model was the most parsimonious with optimal clinical interpretability and class sizes. After consideration of model fit indices, relative class sizes, and interpretability, the five-class model was chosen as the final model (Figure 1). Table 2 displays the class estimate probabilities for the inclusion of each class-defining variable for each class in the final model.

While the seven-class LCA model had more favorable metrics, including model entropy, these gains were minimal compared with the increased complexity of the model. Manual review of this model for face validity revealed almost-identical classes as the five-class LCA model with the addition of two small classes (4.1% and 1.9% of the study population), psychoses and polysubstance use, which were subsumed by other classes in the final five-class model. The mean posterior latent class probabilities for class membership exceeded 0.94 for all classes in the five-class model. The five classes are summarized as the following patient subphenotypes: Chronic pain (class 1); Alcohol use (class 2); Depression with chronic pain (class 3, hereafter “depression with pain”); Psychosis, liver disease, and polysubstance use (class 4); and Pregnancy (class 5).

### 3.2. Subphenotype Characteristics

Descriptive statistics of the patient characteristics for each subphenotype are shown in Table 3. Females constituted the majority of patients in classes 1 (chronic pain), 3 (depression and pain), and 5 (pregnancy). Class 5 (pregnancy) contained the highest proportion of patients located in central metropolitan areas, while class 3 (depression and pain) had the highest proportion located in areas outside of central metropolitan locations. Class 2 (alcohol use) contained the highest proportion of patients designated as self-pay. For between-group comparisons, odds ratios are provided in the multimedia appendices (Appendix A).

### 3.3. Outcomes and Cost Analysis

The results for patient outcomes of hospitalization, in-hospital death, and ED charge are presented in Table 4. Class 1 (chronic pain) was chosen as the reference category as it represented the largest subphenotype. Compared with the reference class, the odds for hospitalization, in-hospital death, and higher ED charges were higher for all other classes. The only exception to this general trend was the pregnancy class (class 5) with regard to the outcome of death, as a very small proportion of patients in the pregnancy class died (OR = 0.00, 95% CI 0.00-0.00). Classes 2 (alcohol use), 3 (depression and pain), and 4 (psychosis, liver disease, and polysubstance use) had the greatest odds of hospital admission, while class 4 carried the greatest odds of in-hospital death. The median ED charge per encounter ranged from USD 2177 to USD 2881, with the lowest and highest charges belonging to class 1 and class 4, respectively.

## 4. Discussion

The ED is a critical setting for the initiation of treatment and harm-reduction interventions for patients with opioid misuse. Identifying subphenotypes of patients with opioid-related encounters is a necessary step towards personalized interventions in this setting. Our analysis, utilizing a nationally representative population of ED patients, provides analytic support for five clinically relevant subphenotypes of opioid-related encounters. These subphenotypes best represented the population of ED patients with opioid-related encounters based on a combination of information criteria-based metrics, class sizes, and clinical interpretability. The high mean posterior latent class probabilities for membership in all classes of the model indicated a strong fit of these classes with the dataset. The subphenotypes detected provide a critical framework for conceptualizing the heterogeneity among ED patients with opioid misuse and carry implications for the development of personalized treatment pathways.

Each subphenotype demonstrated distinct demographic profiles and variations in clinical outcomes. The largest subphenotype consisted of those with an opioid-related diagnosis and chronic pain. This subphenotype had a majority female membership and among the highest proportion of all classes for private insurance (excluding subphenotype 5, consisting of pregnant patients). The outcomes of hospital admission, in-hospital death, and total ED charges were among the lowest for this subphenotype. Addressing opioid misuse within this subphenotype can be difficult as patients may rely on opioids for the relief of chronic pain, and opioids may be the cause of both benefit and harm within individual patients. Additionally, chronic pain is a broad diagnosis affecting patients across demographic groups with multiple, often overlapping causes, which further complicates treatment decisions. Previous estimates of opioid misuse in patients with chronic pain have ranged from 21% to 29% [38]. These estimates, along with the current findings, suggest a role for opioid misuse screening in patients with chronic pain syndromes. The recognition of chronic pain diagnoses as a class-defining variable in the largest subphenotype of opioid-related ED encounters necessitates further exploration of how chronic pain, opioid use, and emergency department utilization are related.

Two opioid-related subphenotypes were defined primarily by the co-occurring diagnosis of psychiatric disorders (depression and psychoses). The associations of opioid misuse with both depression and psychosis have been noted previously, and both conditions are associated with an increased risk for opioid overdose [39]. These classes demonstrated the highest proportion of public insurance and the greatest odds for hospital admission. These classes, in aggregate, account for over a quarter of all opioid-related ED encounters, and their presence underscores the potential opportunities in addressing uncontrolled psychiatric disorders in addition to the risks from opioid misuse during an ED encounter.

The subphenotype defined by psychosis also had the highest incidence of stimulant-related disorders (cocaine and amphetamines) and liver disease. As substance use can induce psychosis, it is unclear whether this finding represents primary psychosis, substance-induced psychosis, hepatic encephalopathy, or a combination of these, further complicating treatment. The risks of opioid misuse and misuse of other substances are increasingly seen as multiplicative. It is therefore unsurprising that this class had the greatest odds of in-hospital death and among the greatest odds of hospitalization. The presence of liver disease as a class-defining variable for this class, as opposed to the class defined by alcohol use, likely speaks to more severe substance use patterns including injection drug use within this class and possibly co-infection with the hepatitis C virus (HCV). Patients with HCV have a disproportionately high prevalence of substance use disorders [40]. The presence of liver disease in this population, along with the emergence of improved treatments for HCV, provides an additional rationale for expanded screening for HCV infection in patients with substance use disorders [41]. While a high prevalence of HCV has been noted in urban Eds and ED-based HCV screening programs have been effective, the presence of HCV screening in this setting is highly variable. The high mortality noted within this class provides an argument for expanding HCV screening programs and programs addressing co-occurring substance misuse within the ED setting. The associations between opioid misuse, psychiatric diagnoses, and polysubstance use are complex and often related to social determinants of health such as housing insecurity, poverty, and race [33,42]. Given this complexity, patients within this subphenotype may particularly benefit from ED-based behavioral approaches in addition to pharmaceutical ones. An understanding of the interplay between these factors is critical for the development of effective public health interventions, which should involve not only those that are opioid-specific but also those which address social and psychiatric needs.

The smallest latent class consisted of opioid-related encounters associated with an alcohol use diagnosis. This class had the highest proportions of male membership, lack of insurance, and membership within the lowest income quartile. Alcohol co-involvement in opioid-involved overdose deaths is common and associated with binge drinking [43]. Given the relatively high odds for hospitalization and in-hospital death within this subphenotype, clinicians should routinely evaluate for problematic alcohol use behaviors in patients with opioid misuse. Effective outpatient interventions for problematic alcohol use should be considered and evaluated within the ED setting. Similarly, policy initiatives in opioid overdose prevention should attempt to address alcohol misuse with accessible intervention programs including resources for those with no insurance or from disadvantaged financial backgrounds.

The identification of the pregnancy subphenotype of opioid-related ED encounters, which represented nearly 12.0% of all opioid-related ED encounters, is consistent with the increased health care engagement and prevalence of opioid use disorder previously described during pregnancy [44]. The incidence of hospital births complicated by opioid use disorder increased more than fourfold from 1999 to 2014 [45]. ED encounters related to pregnancy and opioid use provide opportunities for discussions regarding the treatment options available to prevent both fetal and maternal opioid-related complications during pregnancy. Some of these options have been shown to decrease the severity of neonatal abstinence syndrome and decrease emergency health care utilization in the postpartum period [44,46]. Eds should foster partnerships with clinicians caring for pregnant patients to ensure that appropriate follow-up for both opioid misuse and pregnancy can be assured following discharge from the emergency setting. The detection of opioid misuse in pregnancy represents an opportunity for both the initiation of effective treatments and cost savings for the health care system.

The five-class model has similarities but also important differences when compared with those from other clinical settings. In the hospitalized setting, for example, Liu et al. detected five subphenotypes in their analysis of opioid-related hospitalizations in Pennsylvania [21]. Some subphenotypes were markedly similar to ours, specifically those defined by pregnancy and by polysubstance use with co-occurring psychiatric disorders. However, our model noted additional subphenotypes defined by depression, alcohol use, and chronic pain that were not previously described. These differences are likely due to differences in the variables chosen for analysis and differences between the ED patient population and hospitalized patients. In our analysis, the largest subphenotype of opioid-related ED encounters was defined by chronic pain. This subphenotype likely has overlap with the largest subphenotype identified in hospitalized patients, which was defined by opioid use disorder without co-occurring psychiatric disorders. The recognition of chronic pain diagnoses as a class-defining variable for the largest opioid-misuse subphenotype provides critical context for interventions and barriers to treatment for patients within this subphenotype.

Although not all ED patients with opioid misuse will fall neatly into one of the five subphenotypes identified in this analysis, the study results provide an initial framework through which to understand and tailor care for this patient population. This model may be particularly useful in assisting clinicians in evaluating comorbidities and identifying appropriate pharmaceutical and non-pharmaceutical interventions. This five-class model represents a first step towards the development of personalized treatment approaches for ED patients with opioid misuse and invites further research into the relationship between subphenotypes, specific interventions, and outcomes.

There are several limitations to the current study. The diagnosis codes used as class-defining variables are frequently insensitive and may miss cases of opioid misuse not captured by billing codes. The entry of diagnosis codes by providers requires the detection of a disease process and the entry of the diagnosis into the electronic health record, which contributes to the low sensitivity. Additionally, diagnosis codes do not reliably differentiate between types of opioids or reasons for use. The selection of class-defining variables was based on previous research utilizing LCA in other settings [20,21]. Other potentially informative class-defining variables were possibly excluded from model development. The selected class-defining variables, including chronic pain, are themselves simplifications of complex disease processes and may add some degree of heterogeneity to the subphenotypes described. Individual observations were assigned to a class by their highest posterior probability to facilitate data presentation and interpretation. In practice, however, subphenotypes are not truly mutually exclusive, and patients may have characteristics associated with multiple classes. The models developed were based on nationally representative data. Model validation is needed to ensure that the detected subphenotypes and associated outcomes persist within smaller geographic subdivisions. Last, latent class analysis relies on a combination of the analytic interpretation and evaluation of clinical interpretability; therefore, agreement among subject matter experts may vary.

## 5. Conclusions

This latent class analysis of ED encounters in the United States demonstrated five distinct and clinically relevant subphenotypes of ED patients with opioid-related encounters: Chronic Pain (class 1); Alcohol use (class 2); Depression and pain (class 3); Psychosis, liver disease, and polysubstance use (class 4); and Pregnancy (class 5). It is important to recognize that the characteristics and needs of ED patients with opioid-related encounters may differ greatly. The detection of these subphenotypes should inform treatment considerations, resource allocation, and future research on the efficacy of patient-tailored ED-initiated interventions for opioid misuse.

## Figures and Tables

**Figure 1 ijerph-19-08882-f001:**
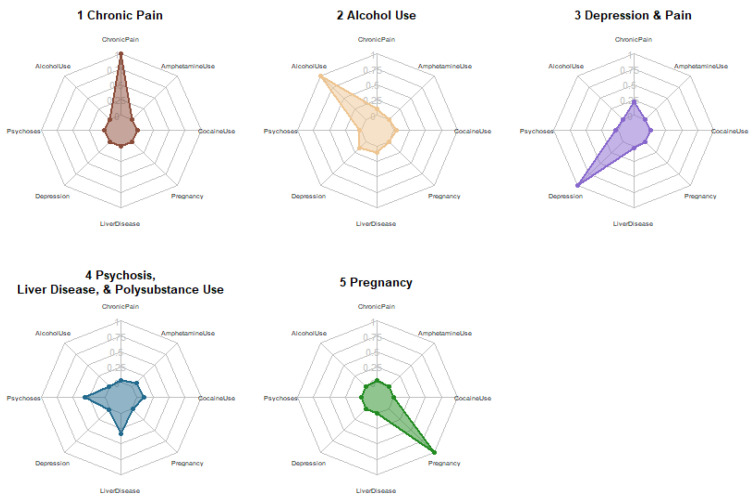
Radar plots for five subphenotypes of ED patients with opioid-related encounters.

**Table 1 ijerph-19-08882-t001:** Information criterion for all models.

Model	AIC	aBIC	χ^2^	Entropy	Smallest Class Size (%)
1 class	83,278,912	83,279,015	16,759,968 (501)	NA	NA
2 class	75,387,096	75,387,312	9,191,275 (492)	0.908	46.6%
3 class	71,463,079	71,462,409	5,424,310 (481)	0.934	11.7%
4 class	69,347,569	68,348,013	3,396,531 (470)	0.948	19.0%
5 class	68,231,307	68,231,865	2,326,259 (459)	0.978	10.6%
6 class	67,675,778	67,676,450	1,796,618 (450)	0.963	4.5%
7 class	67,162,510	67,163,296	1,086,526 (442)	0.990	1.9%
8 class	66,415,227	66,416,127	538,386 (432)	0.952	0.6%
9 class	66,124,289	66,125,302	307,504 (415)	0.968	1.0%
10 class	64,841,155	65,842,282	32,548 (412)	0.980	0.4%

AIC, Akaike information criterion; aBIC, adjusted Bayesian information criterion.

**Table 2 ijerph-19-08882-t002:** Heat map of 5-class LCA model by class estimate of candidate class-defining variables.

	Class 1 Chronic Pain	Class 2 Alcohol Use	Class 3 Depression & Pain	Class 4 Psychosis, Liver Disease & Polysubstance Use	Class 5 Pregnancy
Class size	48.9%	11.0%	17.1%	11.1%	11.9%
**Class-defining variable**					
Chronic pain	1.000	0.104	0.221	0.039	0.038
Alcohol use	0.000	1.000	0.000	0.012	0.000
Psychoses	0.006	0.031	0.034	0.322	0.001
Depression	0.000	0.158	1.000	0.022	0.009
Liver disease	0.010	0.108	0.025	0.329	0.000
Pregnancy	0.002	0.001	0.003	0.003	1.000
Cocaine use	0.002	0.048	0.011	0.103	0.001
Amphetamine use	0.001	0.015	0.007	0.085	0.001

LCA, latent class analysis.

**Table 3 ijerph-19-08882-t003:** Covariate analysis of patient characteristics by latent class.

	Class 1 Chronic Pain	Class 2 Alcohol Use	Class 3 Depression & Pain	Class 4 Psychosis, Liver Disease & Polysubstance Use	Class 5 Pregnancy
n	6,477,223	1,377,526	2,234,701	1,288,114	1,565,534
Age (median, IQR)	48 (32–62)	48 (35–57)	52 (35–67)	47 (32–60)	27 (22–31)
Sex					
Female	59%	30%	67%	43%	100%
Male	41%	70%	33%	57%	0%
Payer					
Medicare	29%	19%	41%	32%	1%
Medicaid	25%	33%	24%	32%	55%
Private	28%	23%	24%	18%	31%
Self-pay	13%	21%	7%	14%	9%
No charge	0%	1%	0%	1%	0%
Other	5%	4%	3%	3%	3%
Median income					
Top quartile	39%	36%	34%	40%	42%
2nd quartile	27%	25%	28%	26%	27%
3rd quartile	20%	21%	22%	20%	19%
4th quartile	14%	19%	17%	14%	12%
Urbanicity					
Central metropolitan	28%	34%	24%	34%	39%
Fringe metropolitan	20%	22%	22%	21%	20%
250–999 K	22%	21%	24%	21%	20%
50–250 K	11%	10%	11%	9%	9%
Micropolitan	11%	8%	12%	9%	8%
Non-core	8%	5%	7%	6%	5%

Note: Discharge weights were used in generating descriptive statistics.

**Table 4 ijerph-19-08882-t004:** Patient outcomes by latent class membership.

Latent Class	Descriptor	Hospital Admission		In-Hospital Death		ED Charges	
		**%**	**OR (95% CI)**	**Count** **(per 1000)**	**OR (95% CI)**	**Median (USD)**	**OR ^a^ (95% CI)**
Class 1	Chronic pain	10.4%	ref.	1.7	ref.	$2177	ref.
Class 2	Alcohol use	32.9%	4.38 (4.36–4.40)	6.5	1.98 (1.95–2.00)	$2817	1.26 (1.26–1.27)
Class 3	Depression & pain	37.0%	5.24 (5.22–5.27)	6.7	2.01 (1.99–2.04)	$2645	1.22 (1.22–1.22)
Class 4	Psychosis, liver disease & polysubstance use	37.1%	5.33 (5.31–5.36)	18.9	3.4 (3.39–3.48)	$2881	1.29 (1.29–1.30)
Class 5	Pregnancy	12.5%	1.24 (1.24–1.25)	<0.1	0.00 (0.00–0.00)	$2605	1.07 (1.07–1.08)

All odds ratios unadjusted; ED, emergency department; OR, odds ratio based on three -step procedure; USD, United States dollar; ref, reference category. ^a^ Natural log transformed.

## Data Availability

Primary data supporting reported results can be obtained from the Healthcare Cost and Utilization Project: Available online: https://www.hcup-us.ahrq.gov/nedsoverview.jsp (accessed on 23 October 2020).

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
