# Peer review of "The Identification of Subphenotypes and Associations with Health Outcomes in Patients with Opioid-Related Emergency Department Encounters Using Latent Class Analysis"

_ijerph, 2022, doi:10.3390/ijerph19148882_

Round 1

Reviewer 1 Report

Dear Authors

The presented manuscript relates to very imprtant issue of opioid-misuse patients and their encounters with emergency departments. The accurate diagnose of the possibble problems and patients' needs will be the key to appropriate treatment- so distinguishing different sub-phenotypes of patients should accelerate the treatment.

In my opinion the research is well designed and the statists is correct.

The results are clearly described in the manuscript.

Whatsoever I do lack one important thing- you have emphasized the importance of these subphenotypes for patients outcomes and choosing potential treatment and intervention pathways (lines 75-78). Unfortunatelly I can not find any informations (in the Discussion and Conclusions chapters) how pointed 5 subphenotypes will change or influence the diagnosis or treatment options (maybe except of HCV screening need) in ED.

As a clinician I think that such recognition between different groups of patients make sens when leading to quicker and better help/treatment 

And as I understend it is one of the most important points of the manuscript- so please address what futre implications will every group bring to the treatment pathway, how it will ease or advance or change the nowadays diagnose and treatment pathways.

Author Response

The presented manuscript relates to very imprtant issue of opioid-misuse patients and their encounters with emergency departments. The accurate diagnose of the possibble problems and patients' needs will be the key to appropriate treatment- so distinguishing different sub-phenotypes of patients should accelerate the treatment. In my opinion the research is well designed and the statists is correct. The results are clearly described in the manuscript.

We thank the reviewer for their thoughtful comments and have addressed the specific critiques below.

Whatsoever I do lack one important thing- you have emphasized the importance of these subphenotypes for patients outcomes and choosing potential treatment and intervention pathways (lines 75-78). Unfortunatelly I can not find any informations (in the Discussion and Conclusions chapters) how pointed 5 subphenotypes will change or influence the diagnosis or treatment options (maybe except of HCV screening need) in ED.

As a clinician I think that such recognition between different groups of patients make sens when leading to quicker and better help/treatment 

And as I understend it is one of the most important points of the manuscript- so please address what futre implications will every group bring to the treatment pathway, how it will ease or advance or change the nowadays diagnose and treatment pathways.

We thank the reviewer for their substantive comments and have, in response, expanded the discussion in sections related to specific subphenotypes to provide more concrete examples of how these subphenotypes should change or influence diagnostic and/or treatment options. In addition to the section on HCV screening (269-276) as pointed out by the reviewer, examples of changes to address these concerns are on lines 250-251, 294-295, and 308-310.

Reviewer 2 Report

The study provides an approach to recognize the characteristics and needs of ED patients with opioid-related encounters. The results are significant and interesting. I raise main comments below:   

In Table 4, the results showed the association between classes and outcomes with unadjusted OR. Instead of the traditional methods with 5 classes LCA, why did the authors not consider analyzing the effect of hospitalization/mortality caused by each demographic covariate and class-defining variable directly? The exploration of multi-variates’ relationships with the outcomes can inform readers more evidently.

The reason of individual opioid using may change over time. In other words, patients may use/misuse drug for different purpose in different time. How did the authors deal with the problem?

Minor comments

Using the term ‘subphenotype needs to be take into consider. In general, phenotype may easily be linked to the physical expression of DNA.

In Table 3, the statistics test for a difference is needed.

In Line 152, the statement ‘they represent associations between covariate and class and were not adjusted for other covariates’ needs to be elaborate more.

The definition of the strata of urban-rural designation needs to be added.

Line 237, the authors addressed that class 1 subphenotype had a majority female membership and among the highest proportion of all classes for private insurance. Did they neglect the class 5?

Table s3 is redundant.

The significance or confidence interval of OR and notation should be added in Table s4 and discuss more.

Author Response

Reviewer 2

The study provides an approach to recognize the characteristics and needs of ED patients with opioid-related encounters. The results are significant and interesting. I raise main comments below:   

In Table 4, the results showed the association between classes and outcomes with unadjusted OR. Instead of the traditional methods with 5 classes LCA, why did the authors not consider analyzing the effect of hospitalization/mortality caused by each demographic covariate and class-defining variable directly? The exploration of multi-variates’ relationships with the outcomes can inform readers more evidently.

We thank the reviewer for their thoughtful comment and suggestion. We chose not to present the associations of each demographic covariate and class-defining variable separately as the aim of the analysis was to identify and describe the subphenotypes. The outcomes presented in table 4 serve to highlight that each of these subphenotypes has important distinctions in these outcomes. Presenting each demographic feature and variable that was used in identifying the subphenotypes would be too cumbersome for a manuscript table and would likely confuse readers.

The reason of individual opioid using may change over time. In other words, patients may use/misuse drug for different purpose in different time. How did the authors deal with the problem?

We agree with the reviewer that reasons for drug use may change over time. For this reason, we were thoughtful to limit our analysis to the detection of subphenotypes by patient characteristic rather than opioid-specific encounter reason (which were likely to change over time). Additionally, each encounter was included as a separate observation even though some likely included repeat encounters for the same patient at different time points. We clarified this rationale in lines 97 through 100.

 Minor comments

Using the term ‘subphenotype’ needs to be take into consider. In general, phenotype may easily be linked to the physical expression of DNA.

This definition is an important one. We utilized the definition employed by other studies evaluating subphenotypes for different disease processes as a subgroup among a disease entity with either varying risks for poor outcomes (prognostic enrichment) or sharing underlying biologic factors or a different reaction to medical measures (predictive enrichment). In this case, we highlighted important varying risks for poor outcome. PMIDs: 33478589, 32526190. A sentence was added at line 60 to highlight this point.

In Table 3, the statistics test for a difference is needed.

We state in line 155 that “due to the large sample size, traditional significance tests and p-values were uninformative (all p-values <0.001) and standard errors were generally < 0.001. Empahsis was thus placed on descriptive statistics and odds ratios for class membership or class makeup based on covariates.” We therefore decided against providing uninformative direct tests for the table.

In Line 152, the statement ‘they represent associations between covariate and class and were not adjusted for other covariates’ needs to be elaborate more.

We added phrases the sentence to make the analysis more clear.

The definition of the strata of urban-rural designation needs to be added.

This is clarified in the methods section lines 121-123.

Line 237, the authors addressed that class 1 subphenotype had a majority female membership and among the highest proportion of all classes for private insurance. Did they neglect the class 5?

A clarifying point was added to the sentence to account for this.

Table s3 is redundant.

Table S3 provides the STROBE checklist for standardized reporting.

The significance or confidence interval of OR and notation should be added in Table s4 and discuss more.

These findings are discussed in relation to table 4 and, as above, CIs and significant with p-values were excluded for reasons as addressed in the methods section.

Round 2

Reviewer 2 Report

I have no more comment.